# Targeted MicroRNA Profiling Reveals That Exendin-4 Modulates the Expression of Several MicroRNAs to Reduce Steatosis in HepG2 Cells

**DOI:** 10.3390/ijms241411606

**Published:** 2023-07-18

**Authors:** Olfa Khalifa, Khalid Ouararhni, Khaoula Errafii, Nehad M. Alajez, Abdelilah Arredouani

**Affiliations:** 1Diabetes Research Center, Qatar Biomedical Research Institute, Hamad Bin Khalifa University, Qatar Foundation, Doha P.O. Box 34110, Qatar; 2Genomics Core Facility, Qatar Biomedical Research Institute, Hamad Bin Khalifa University, Qatar Foundation, Doha P.O. Box 34110, Qatar; 3African Genome Center, Mohammed VI Polytechnic University (UM6P), Ben Guerir 43151, Morocco; 4Translational Cancer and Immunity Center, Qatar Biomedical Research Institute, Hamad Bin Khalifa University, Qatar Foundation, Doha P.O. Box 34110, Qatar; 5College of Health & Life Sciences, Hamad Bin Khalifa University, Qatar Foundation, Doha P.O. Box 34110, Qatar

**Keywords:** steatosis, NAFLD, exendin-4, miRNAs, HepG2, GLP-1R agonist

## Abstract

Excess hepatic lipid accumulation is the hallmark of non-alcoholic fatty liver disease (NAFLD), for which no medication is currently approved. However, glucagon-like peptide-1 receptor agonists (GLP-1RAs), already approved for treating type 2 diabetes, have lately emerged as possible treatments. Herein we aim to investigate how the GLP-1RA exendin-4 (Ex-4) affects the microRNA (miRNAs) expression profile using an in vitro model of steatosis. Total RNA, including miRNAs, was isolated from control, steatotic, and Ex-4-treated steatotic cells and used for probing a panel of 799 highly curated miRNAs using NanoString technology. Enrichment pathway analysis was used to find the signaling pathways and cellular functions associated with the differentially expressed miRNAs. Our data shows that Ex-4 reversed the expression of a set of miRNAs. Functional enrichment analysis highlighted many relevant signaling pathways and cellular functions enriched in the differentially expressed miRNAs, including hepatic fibrosis, insulin receptor, PPAR, Wnt/β-Catenin, VEGF, and mTOR receptor signaling pathways, fibrosis of the liver, cirrhosis of the liver, proliferation of hepatic stellate cells, diabetes mellitus, glucose metabolism disorder and proliferation of liver cells. Our findings suggest that miRNAs may play essential roles in the processes driving steatosis reduction in response to GLP-1R agonists, which warrants further functional investigation.

## 1. Introduction

Non-alcoholic fatty liver disease (NAFLD) is a clinicopathologic illness defined by excessive fat accumulation in the liver due to causes other than excessive alcohol use or viral infection. This illness encompasses simple steatosis (benign fatty infiltration), non-alcoholic steatohepatitis (NASH) (fatty infiltration with inflammation), fibrosis, and cirrhosis, which can develop into hepatocellular cancer [1,2,3]. NAFLD is linked to insulin resistance and genetic vulnerability [4], and because of the rising incidence of obesity and obesity-related metabolic syndrome, NAFLD has become the primary cause of chronic liver disease in industrialized nations and the third cause of liver transplantation [5,6,7,8]. Therefore, NAFLD is a significant public health issue [9] but presently has no approved pharmacotherapy. Sustained weight loss of at least 5% of the total body weight is beneficial for NAFLD patients, as reflected by improved liver enzyme levels and reduced liver fat content [10,11,12]. A loss of more than 10% of body weight appears to minimize inflammation and harm to liver cells and may even repair some fibrosis damage [13,14,15]. Nevertheless, most people find it challenging to achieve the weight loss required to improve their NAFLD and much harder to maintain their weight loss [16]. Consequently, there is a critical unmet medical need for effective pharmacological treatments for NAFLD that are not reliant on weight loss.

Glucagon-like peptide-1 receptor agonists (GLP-1RAs), which are modified variants of the endogenous GLP-1, such as exendin-4 (Ex-4), reduce blood glucose levels and induce weight reduction, and are hence an effective therapeutic option for type 2 diabetes mellitus (T2DM) [17]. Current research shows that GLP-1RAs can lower hepatic lipid storage significantly by inhibiting fatty acid synthesis-related genes and promoting fatty acid oxidation-related expression levels [18,19]. However, the effect of GLP-1RAs on the miRNA landscape in hepatic steatosis is yet unknown. MicroRNAs (miRNAs) are 22 nt regulatory RNA molecules that regulate gene expression post-transcriptionally. Since their discovery in *Caenorhabditis elegans* in 1993 [20], these short non-coding RNAs have been linked to various biological processes, including development, metabolic regulation, aging, and disease progression [21]. Notably, miRNAs have recently been found to play a significant role in lipid metabolism, inflammation, cell death, and tissue development, all of which significantly contribute to the risk of NAFLD [22,23]. Because of their excellent stability in peripheral blood, miRNAs may be employed as diagnostic or predictive biomarkers for various human illnesses. Some of the miRNAs that were reported to be implicated in NAFLD include miR-219a, miR-373, miR-378c, miR-590, miR-3611, miR-376b, miR-186, miR-17, miR-1286, miR-5699, miR-183, miR-31, miR-150, miR-182, miR-200a, miR-224, miR-92b, miR-3613, miR-708, and miR-766 in humans [24]; miR-126, miR-150, miR-223, miR-483-3p, miR-1226, and miR-1290 in HepG2 cells [25]; and miR-351, miR-434, miR-467a, and miR-682 in mice [26]. 

In the current study, we established an in vitro hepatic steatosis model by treating HepG2 with oleic acid (OA). Typically, OA is used to establish in vitro steatosis models in different cell types, including HepG2 [27] and GLP-1RA exendin-4 (Ex-4), and is used to reduce lipid accumulation. We used NanoString technology to profile 799 highly curated human miRNAs from miRBase 22 in control, steatotic, and Ex-4-treated steatotic cells. Under the same condition, we validated a subset of differentially expressed miRNAs using qRT-PCR. Previously we performed transcriptomics analysis [28] and used the mRNA data to identify the gene targets of the differentially expressed miRNAs.

## 2. Results

### 2.1. Study Design

We cultured HepG2 cells in 6-well plates until there was 70% confluency and then starved them for several hours in DMEM containing 1% fatty-acid-free FBS. Following starvation, we incubated the cells at 37 °C for 16 h with 1% FBS DMEM containing 400 µM OA solution to induce steatosis. Following steatosis induction, we incubated the cells for three hours in fresh 1% FBS DMEM containing 400 µM OA solution in the absence or presence of 200 nM Ex-4. After treatment, we extracted the total RNA, including miRNA. The samples were analyzed using NanoString technology for a miRNA panel which included 799 highly curated human miRNAs, to identify differentially regulated miRNAs between different treatments. Finally, the data were examined using various bioinformatics tools (Figure 1); Figure 1 was prepared using BioRender.

### 2.2. Exendin-4 Reduces OA-Induced Lipid Accumulation in HepG2

In agreement with our previous publications [18,28,29], treatment of HepG2 cells with 400 mM OA induced a significant accumulation of TGs in lipid droplets, which we could easily quantify or visualize with BODIPY staining. Treatment of the steatotic cells with 200 nM Ex-4 significantly reduced the lipid accumulation, as reflected by the lowering of TG content, the reduction of BODIPY staining, and the downregulation of the expression of perilipin mRNAs.

### 2.3. Identification of Differentially Expressed miRNAs

After normalization and filtering, 101, 105, and 82 miRNAs were identified in control, steatotic, and Ex-4-treated cells, respectively. The list of all the miRNAs is shown in Appendix A; of these miRNAs, 78 are shared by the three groups, 21, 2, and 1 miRNA were identified commonly between each pair of groups, i.e., control versus steatotic, steatotic versus Ex-4-treated, and control versus Ex-4-treated cells, while 1, 4, and 1 miRNA were expressed exclusively in control, steatotic, and Ex-4-teared cells, respectively (Appendix A). The hierarchical clustering heatmap in Appendix A visualizes the distinct miRNA profiles between the different treatment conditions using all the miRNAs. To better understand the role of miRNAs in steatosis and the Ex-4 beneficial effect on steatosis, we compared (a) steatotic cells versus control cells and (b) Ex-4-treated cells versus steatotic cells. The results of each comparison are presented separately for convenience in Sections (a) and (b) below:

(a)Differentially expressed miRNAs in steatotic versus control cells

When comparing the miRNA expression profile of steatotic and control cells, 99 miRNAs were shared between the two groups, while 2 and 6 miRNAs were expressed exclusively in control cells and steatotic cells, respectively (Figure 2A). We detected nine differentially expressed miRNAs (DEmiRs) (FC > 2 and FDR < 0.05) between control and steatotic cells, with two being downregulated (hsa-miR-30e-p and hsa-miR-1246), and seven being upregulated (hsa-miR-345-5p, hsa-miR-379-5p, hsa-miR-362-5p, hsa-miR-651-5p, hsa-miR-421, hsa-miR-122-5p, and hsa-miR-4488) in steatotic compared to control cells (Figure 2A–C). The hierarchical clustering heatmap based on the *t*-test (*p* < 0.05) in Figure 2D visualizes the distinct miRNA profiles between steatotic and control cells using the top 25 DEmiRs.

Chemometric analysis

In addition to the differential expression analysis, we performed principal component analysis (PCA) on the data sets from the control and steatotic cells (Figure 3). The scores plot in Figure 3A shows a clear separation between the two groups. The first and second principal components, PC1 and PC2, accounted for 37.1% and 20.7% of the total variance in the data. We further performed partial least squares discriminant analysis (PLS-DA) to identify the most critical miRNAs to separate the two groups. In the PLS-DA scores plot in Figure 3B, PC1 and PC2 account for 33.2% and 24.3% of the total variance, respectively. Using leave out one cross-validation (LOOCV), the R2 and Q2 values of the model were, respectively, 0.92 and 0.64 for PC1 and 0.99 and 0.78 for PC2, suggesting that the model is robust. We then used the variable important in projection (VIP) score to identify the essential miRNAs for the separation that we see in the PLS-DA score plot. We identified 42 miRNAs that have a VIP > 1 (the threshold that is typically used to select relevant features) (Appendix A), and Figure 3C shows the top 15 features that contribute the most to the separation we see in Figure 3B. 

Pathway enrichment analysis

To investigate the signaling pathways that could potentially implicate the DEmiRs between the steatotic and control cells, we first identified their bona fide mRNA targets. Hence, we uploaded the DEmiRs into the Ingenuity Pathway Analysis application, along with the DEmRNAs (differentially expressed mRNAs) between the two groups, which we obtained by whole transcriptome profiling under the same experimental conditions, as used in our previous paper [28]. We identified the mRNA targets of our DEmiRs using the predicted miRNA–mRNA binding relationships from TargetScan with high confidence, and we only selected the miRNAs-mRNA pairs with opposite directions of expression, i.e., when the miRNA is upregulated, the mRNA is downregulated, and vice versa. Under these conditions, we identified 15 unique gene targets (Table 1). We subsequently used the 15 identified target genes to determine the enriched canonical pathways, diseases, and cellular and molecular functions (Figure 4). Our analysis identified several canonical pathways that were enriched in the DEmiRs target proteins and that were relevant to steatosis and NAFLD, including HIF1a, mTOR, TGF-b, PTEN, PPAR, NF-kB, insulin receptor, PI3K/AKT, hepatic fibrosis /hepatic stellate cell activation, and hepatic fibrosis signaling pathways (Figure 4A,B).

On the other hand, we identified several diseases and cellular functions that are associated with the DEmiR target and might be relevant to NAFLD, including carbohydrate metabolism, hepatic system disease, cell signaling, molecular transport, lipid metabolism, small molecule biochemistry, cellular function and maintenance, endocrine system development and function, inflammatory response, cellular growth and proliferation, inflammatory disease, cell death and survival, endocrine system disorders, and metabolic disease, diabetes mellitus, inflammation of the liver, proliferation of hepatocytes, and cell death of hepatocytes (Figure 4C,D).

(b)The differentially expressed miRNAs in Ex-4-treated versus steatotic cells

When we compared the miRNA datasets from the Ex-4-treated and steatotic cells, 80 miRNAs were shared between the two groups, while 25 and 2 miRNAs were expressed, respectively, only in the Ex-4-treated and steatotic cells (Figure 5A). We detected 34 differentially expressed miRNAs (DEmiRs) (FC > two and FDR < 0.05) between Ex-4-treated and steatotic cells, with 2 being upregulated and 32 being downregulated in the steatotic compared to control cells (Figure 5B,C). The hierarchical clustering heatmap based on the *t*-test (*p* < 0.05) in Figure 4D illustrates the distinct miRNA profiles between Ex-4-treated and steatotic cells using the top 25 DEmiRs. Notably, there is no miRNA:mRNA pairs that overlap between the steatotic vs controls cells and Ex-4-treated vs steatotic cells. The only miRNA whose expression was reversed by Ex-4 treatment was hsa-miR-345-5p.

Chemometric analysis

In addition to the differential expression analysis, we performed principal component analysis on the data sets from Ex-4-treated and steatotic cells. The scores plot in Figure 3D shows a clear separation between the two groups. The first and second principal components account for 66.7% and 10% of the total variance in the data, respectively. We further performed PLS-DA to identify the most important miRNAs for separating the two groups. In the PLS-DA scores plot shown in Figure 3E, PC1 and PC2 account for 66.6% and 10% of the total variance, respectively. Using LOOCV, the R2 and Q2 values of the model were, respectively, 0.91 and 0.83 for PC1 and 0.99 and 0.90 for PC2, indicating the robustness of the model. We then used the variable important in projection (VIP) score to identify the most important miRNAs for the separation we see in the PLS-DA scores plot. We identified 57 miRNAs that have VIP > 1 (the threshold that is typically used to select relevant features) (Appendix A), and Figure 3F shows the top 15 features that contribute the most to the separation we see in Figure 3. The raw data are available in Appendix A.

Pathway enrichment analysis

To explore the signaling pathways associated with the DEmiRs between Ex-4-treated and steatotic cells, we first identified their mRNA targets. To do so, we uploaded the DEmiRs into the IPA application, along with the DEmRNAs between the two groups, which we obtained by whole transcriptome profiling under the same experimental conditions used in our previous paper [28]. We identified the mRNA targets of our DEmiRs using the predicted miRNA–mRNA binding relationships from TargetScan with high confidence. We only selected miRNA–mRNA pairs with opposite directions of expression, i.e., when the miRNA was upregulated, the mRNA was downregulated, and vice versa. Under the conditions, we identified 34 unique gene targets (Table 2). 

We also identified several diseases and cellular functions that are associated with the DEmiRs target and that might be relevant to NAFLD, including carbohydrate metabolism, hepatic system disease, lipid metabolism, small molecule biochemistry, endocrine system development and function, inflammatory response, inflammatory disease, cell death and survival, endocrine system disorders, and metabolic disease (Figure 4G,H).

To further explore and annotate the potential molecular pathways affected by Ex-4-treated, we used the experimentally validated DEmiRs gene targets to perform the protein–protein interaction (PPI) analysis using the STRING tool [30] (https://string-db.org/ accessed on 10 January 2023). The PPI network showed several significant (FDR < 0.05) enrichments and associations of the multiple proteins in mediating several pathways (Figure 6A). There were 93 significant (FDR < 0.05) Gene Ontology terms for the biological processes. The top ten are shown in Figure 6B and include extracellular matrix organization, animal organ morphogenesis, blood vessel development, and others. On the other hand, there were five significant GO-terms for molecular, including extracellular matrix structural constituent conferring tensile strength, extracellular matrix structural constituent, growth factor binding, platelet-derived growth factor binding, and β-catenin binding. Finally, there were eleven significant GO-terms for cellular components, including a complex of collagen trimers, collagen trimer, endoplasmic reticulum lumen, fibrillar collagen trimer, extracellular matrix, endoplasmic reticulum, basement membrane, collagen type IV trimer, plasma membrane protein complex, receptor complex, and membrane protein complex.

### 2.4. Validation of miRNA Differential Expression with qRT-PCR

The expression levels of six differentially expressed miRNAs between steatotic and Ex-4-treated cells were validated with qRT-PCR in independent samples. Except miR-1246 (FC = 1.05; −0.01), the expression levels of miR-122-5p (FC = 0.98; −0.078), miR-4488 (FC = 1.05; −0.17), miR-651(FC = 1.01; −0.004), miR-345 (FC = 1.009; −0.011), and miR-379 (FC = 1.009; −0.011), in steatotic cells and Ex-4 treated steatotic cells respectively) were consistent with the NanoString data (Figure 7).

## 3. Discussion

Because of the increasing prevalence of obesity, NAFLD has become one of the most common causes of chronic liver disease [31]; however, NAFLD presently has no approved pharmacotherapy. Nevertheless, recent human and animal research has shown some beneficial effects of the GLP-1R agonists [32,33,34,35,36], but the mechanisms underlying this positive effect remain elusive.

We profiled a panel of 799 highly curated human miRNAs in this investigation to see if the regulation of miRNA expression and gene targets, as well as the associated signaling pathways and biological processes, might explain the observed Ex-4-induced reduction of OA-induced steatosis in HepG2 cells. We identified significant differences in the expression of several miRNAs between the control and steatotic cells on the one hand and between steatotic cells and Ex-4-treated steatotic cells on the other hand. We also found that several of the DemiRs’ target proteins are involved in various biological processes and signaling pathways germane to NAFLD.

Because of the ubiquitous expression of the GLP-1R, the agonists of this receptor exhibit pleiotropic effects in vivo [37]. Treatment with GLP-1R agonists, among other things, results in significant weight loss and enhanced insulin sensitivity, which eventually contribute to a reduction in liver fat content [37]. Recent research, however, suggests that the beneficial effects of GLP-1R agonists on NAFLD may be mediated by direct activation of the GLP-1R on hepatocytes, independent of weight loss [38,39]. Nevertheless, other studies have contested this hypothesis because they could not detect GLP-1R in liver cells due to very low levels of hepatic GLP-1R expression [40]. The GLP-1R expression in the liver cells remains controversial, but several groups, including us, have detected it in HepG2 and human hepatocytes [18,38,39].

Mammalian miRNAs regulate gene expression post-transcriptionally, which impacts signaling pathways involved in many disorders, including NAFLD [23,29,41,42,43,44].

Compared to control cells, steatotic cells exhibited significant up- and downregulation of seven and two miRNAs, respectively (Appendix A). On the other hand, 2 and 34 miRNAs were up- and downregulated in Ex-4-treated steatotic cells compared to steatotic cells, respectively (Figure 4C). The function of several DEmiRs in our study is unknown, whereas many others have been linked to different diseases, including liver disease [45,46]. Remarkably, the expression of six miRNAs (hsa-miR-122-5p, hsa-miR-345-5p, hsa-miR-379-5p, hsa-miR-651-5p, hsa-miR-1246, and hsa-miR-4488), which were upregulated in response to OA exposure, was reversed following Ex-4 treatment. This shift in expression suggests that these five miRNAs may be essential in reducing OA-induced lipid buildup produced by Ex-4 therapy.

Hsa-miR-122-5p is one of the most investigated miRNAs, and multiple studies have revealed that it is involved in various physiological processes of the liver. For instance, miR-122 promotes hepatic lipogenesis by inhibiting the LKB1/AMPK pathway by targeting Sirt1 in NAFLD [47]. A recent study reported that the downregulation of miR-122-5p activates glycolysis via PKM2 in the Kupffer cells of rat and mouse models of NASH [48]. Additionally, miR-122-5p inhibition improves inflammation and oxidative stress damage in dietary-induced NAFLD by targeting FOXO3 [49]. Interestingly, the hepatic and serum miR-122 levels were significantly higher in hepatic steatosis and fibrosis in humans [50,51]. Our network and pathway analysis also indicate that miR-122-5p is associated with NAFLD and diabetes mellitus (Figure 3D). We also found that the miR-122-5p targets MASP1 (mannan-binding lectin serine protease 1). MAPS1 functions as a component of the lectin pathway of complement activation, which plays an essential role in the innate and adaptive immune response. When abnormally activated, the complement system can induce inflammation and damage to normal tissues and participate in the development and progression of various diseases. We could not find any link between MASP1 and NAFLD. However, recent studies have shown that complement activation is involved in the genesis and development of alcoholic liver disease (ALD) [52]. Further investigation is warranted to examine the potential link between miR-122-5p/MASP1 and NAFLD development.

Hsa-miR-345-5p was notably upregulated in steatotic cells compared to control cells in our study with Nanostring and q-PCR (FC = 5.99, FDR = 9.9 × 10^−8^); (FC = 1.009), respectively. Remarkably, Ex-4 treatment reversed the direction of the expression of miR-345-5p to be downregulated in Ex-4 treated cells compared to steatotic cells (FC = 0.16703, FDR = 1.7632 × 10^−6^). This result was validated with q-PCR (FC = −0.011). MiR-345-5p was recently shown to be downregulated in liver fibrosis and prevented the progression of liver fibrosis by suppressing hypoxia-inducible factor-1alpha (HIF1α) expression in mice [53]. The same study showed that the miR-345-5p–HIF1α axis might be a potential therapeutic target for liver fibrosis [53]. Our network analysis indicates that miR-345-5p targets RPS21, which is involved in the mTOR canonical pathway. The mTOR pathway is implicated in developing NAFLD [54,55]. MiR-345-5p also targets SAT1, which is implicated in HIF1a signaling that, in turn, is linked to the development of NAFLD [56,57,58].

Compared to control cells, the upregulation of hsa-miR-379-5p in steatotic cells was also reversed after Ex-4 treatment, suggesting its potential relevance for the beneficial effect of Ex-4-on steatosis in HepG2 cells. Several studies have indicated the role of miR-379 in metabolic pathways. For instance, individuals with early-stage NAFLD have increased serum miR-379-5p expression, implying that it might be used as a biomarker to distinguish NAFLD patients from controls [59]. The same study suggests that miR-379 increases cholesterol lipotoxicity, which promotes the development and progression of NAFLD by interfering with the expression of target genes, including those in the IGF-1 signaling pathway [59]. In addition, hepatic miR-379-5p deficiency reduces serum very-low-density lipoprotein-associated triglyceride (VLDL-TG) levels by promoting hepatic lipid re-uptake and TG accumulation [60]. Recently, Cao and coworkers showed that the lack of miR-379/miR-544 clusters resists high-fat diet-induced obesity and prevents hepatic triglyceride accumulation in mice [61]. Dong et al. lately reported that miR-379-5p inhibits STAT1 expression and regulates cholesterol metabolism through the STAT1/HMGCS1 axis in db/db mice, suggesting miR-379-5p might be applied to improve lipotoxicity and relieve diet induced-liver damage [62]. Altogether, these data indicate that miR-379-5p might play a vital role in the positive effect of Ex-4 on lipid accumulation that we observed in our study.

Hsa-miR-651-5p expression was also upregulated in steatotic cells and became downregulated after Ex-4 treatment, suggesting a potential role of this miR in the beneficial effect of Ex-4 on OA-induced lipid accumulation. These data were validated using q-PCR (FC = 1.01; −0.004) in steatotic cells and Ex-4 treated steatotic cells, respectively. However, the role of miR-651-5p in liver metabolism, steatosis, or NAFLD has not been reported previously, and further investigations are warranted to examine it.

Another miR whose expression was downregulated in steatotic cells and upregulated following Ex-4 treatment in our study was hsa-miR-1246. These results contradicted the q-PCR data where the Ex-4 treatment downregulated hsa-miR-1246 expression. We could not find any studies linking hsa-miR-1246 to steatosis or NAFLD. However, our network and pathway analysis showed that miR-1246 targets CCNG2. CCNG2 is associated with the HIF1a canonical pathway, which is known to be involved in NAFLD [56,57,58,63,64,65].

Hsa-miR-4488 was also upregulated in steatotic cells and downregulated following Ex-4 treatment. The same finding was obtained with q-PCR (FC = 1.05; −0.17) in steatotic cells and Ex-4 treated steatotic cells, respectively. However, we could not locate any study that indicated a relationship between this miR and lipid metabolism, steatosis, or NAFLD. Further studies are, therefore, needed to understand better the role that this miR might play in the beneficial effect of Ex-4.

Ex-4 not only reversed the effect of OA on the expression of the six miRs indicated above, but it also modulated the expression of several other miRs compared to steatotic cells. IPA revealed that many of these miRs target genes linked to canonical pathways or biological functions germane to NAFLD. For instance, hsa-miR-let7-5p targets ACVR1C, AMT, COL1A1, COL27A1, DUSP16, MT-ND4L, TGFBR3, and WNT9A genes. According to IPA, the COL1A1 gene is linked to “fibrosis of the liver”, “cirrhosis of the liver”, “proliferation of hepatic stellate cells”, “diabetes mellitus”, “glucose metabolism disorder”, and “proliferation of liver cells”. The AMT gene is associated with “hepatic steatosis” and “NAFLD”, and the canonical pathway’s “PI3K/AKT signaling”. In contrast, the ACVR1C gene is linked to “hepatic steatosis”, “glucose metabolism disorder”, and the canonical pathway’s “TGF-b signaling”, “WNT/b-catenin signaling”, and “PPARa/RXRa activation pathway”. WNT9A, TGFBR3, and ACVR1C are linked to the canonical pathway’s “WNT/b-catenin signaling” and “HOTAIR regulatory pathway”, known for their implication in NAFLD [18,66,67]. Hsa-miR-4532 targets the RASD2 gene, which is associated with the canonical pathway’s “insulin receptor signaling”, “TGF-b signaling”, “PPARa/RXRa activation pathway”, and “NF-kB signaling”, all of which have known associations with NAFLD [68,69,70,71,72]. Likewise, miR-1237-5p targets PTCH1, ITGA5, and LRP1 genes, which are associated with “cirrhosis of the liver”, “fibrosis of the liver”, “glucose metabolism disorder”, and “production of reactive oxygen species in the liver”, and with the NAFLD-related canonical pathway’s “hepatic fibrosis signaling pathway”, “PI3K/AKT signaling”, and “PTEN signaling” [73]. Interestingly, a recent study reported that the GLP-1R agonist liraglutide ameliorates NAFLD in diabetic mice via the IRS2/PI3K/AKT signaling pathways [74]. Genetic and molecular studies, particularly in the context of non-alcoholic fatty liver disease (NAFLD), support a critical role for PTEN in hepatic insulin sensitivity and the development of steatosis, steatohepatitis, and fibrosis [75]. Additionally, miR21-5p and miR-96-5p target SMAD7, which is linked to “fibrosis of the liver” [76], “proliferation of liver cells”, “proliferation of hepatocytes”, “diabetes mellitus”, and “glucose metabolism disorder”. SMAD7 is also associated with the canonical pathway’s “TGF-b signaling”, “hepatic fibrosis signaling pathway”, and “hepatic fibrosis/hepatic stellate cell activation”. 

In recent years, the role of liver-resident cells, such as Kupffer cells and hepatic stellate cells (HSCs), in the development of NAFLD has been implicated. Kupffer cells are specialized macrophages that exist in the liver and play an important role in liver homeostasis. Kupffer cells are activated in NAFLD, resulting in an increase in the production of pro-inflammatory cytokines and chemokines, which contribute to the development of hepatic inflammation and fibrosis [77]. HSCs are another type of liver-resident cell that plays an important role in the pathophysiology of NAFLD. They are found in the Disse area and are in charge of vitamin A storage as well as the generation of extracellular matrix proteins. HSCs become activated in response to liver injury or inflammation, resulting in their metamorphosis into myofibroblast-like cells. These cells are characterized by their ability to produce excessive amounts of extracellular matrix proteins, which contribute to the development of liver fibrosis [78]. Several studies have shown that Kupffer cells and HSCs have a role in the pathophysiology of NAFLD. Miura et al., for example, discovered that Kupffer cells play a critical role in the development of steatohepatitis in mice fed with a high-fat diet. The authors discovered that removing Kupffer cells from these mice reduced hepatic inflammation and fibrosis [79]. Marra et al. discovered that HSCs are activated in the livers of NAFLD patients and that this activation is related to the severity of liver fibrosis [80]. Given that different microRNAs targeting inflammation and liver fibrosis were altered in the steatotic and Ex-4 treated steatotic HepG2 cells, one cannot exclude a potential in vivo effect of the GLP-1R agonists on the miRNAs profiles in Kupffer cells and HSCs to lower inflammation and liver fibrosis and thus improve NAFLD. Future investigations are warranted to investigate this hypothesis.

Our study revealed that specific miRNAs are up or downregulated in the steatotic HepG2 cells compared to control cells, whereas the Ex-4 treatment of steatotic cells affected the expression of additional miRNAs compared to steatotic HepG2 cells. This observation underscores the potential role of these miRNAs in the modulation of the expression of a myriad of genes involved in hepatic lipid metabolism and inflammation, which ultimately leads to improvement of steatosis in the present study, and NAFLD in vivo. Our study paves the way for future in vivo studies to better understand the contribution of the modulation of the miRNA profile of hepatocytes, and maybe other liver cells, to the positive effect of GLP-1R agonists on NAFLD. The understanding of the full mechanisms whereby each miRNA contributes to the reduction of lipid accumulation upon Ex-4 treatment will require further investigation and may open new avenues for the discovery of new drug targets for NAFLD.

Overall, our findings show that the Ex-4 cell treatment simultaneously affects the activity of numerous steatosis-related signaling pathways by modulating the expression of distinct miRNAs, which may explain the observed significant reduction in lipid accumulation.

## 4. Material and Methods

### 4.1. HepG2 Culture and OA Preparation

The human hepatoma HepG2 cell line (HB-8065, ATCC) was obtained from ATCC (Manassas, VA, USA) and was cultured in Dulbecco’s modified Eagle’s medium (DMEM) (31966047, Gibco, Waltham, MA, USA) supplemented with 10% FBS (10500064, Gibco, MA, USA) and 1% penicillin/streptomycin (15070063, Gibco, MA, USA) at 37 °C and 5% CO_2_. We carried out all the experiments with cells passaged no more than 25 times. We prepared the oleic acid (OA) solution as described in [81]. Briefly, OA (O-1008 Sigma-Aldrich, Darmstadt, Germany) powder was dissolved at a final concentration of 12 mM in phosphate-buffered saline (PBS; 137 mM NaCl, 10 mM phosphate, 2.7 mM KCl, and pH 7.4) containing 11% fatty acid-free bovine serum albumin (FFA-BSA; 0215240110, MP Biomedicals, Santa Ana, CA, USA). The solution was then sonicated and shaken overnight at 37 °C with an OM10 orbital shaking incubator (Ratek Instruments Pty, Ltd., Boronia, Australia). The OA solution was filtered with a 0.22 μM filter, aliquoted, and kept at 4 °C. We utilized a fresh aliquot for each experiment.

### 4.2. Induction of Steatosis and Treatment with Exendin-4

To create the steatosis cell model and treat it with Ex-4, we used the same procedure as in our recent publications [18,28]. In brief, we cultured HepG2 cells in 6-well plates at a density of 4 × 10^5^ cells/well until 70% confluency was reached, then starved them for 6 h in DMEM containing 1% fatty-acid-free FBS. Following starvation, we incubated the cells for 16 h at 37 °C in DMEM containing 400 mM OA and 1% fatty-acid-free FBS and then quantified steatosis. We used 1% fatty-acid-free FBS in all OA treatment experiments to ensure that OA was the single inducer in the medium and that OA did not react with components of FBS. Following steatosis induction, we washed the cells and incubated them for three hours in fresh 1% FBS DMEM containing 400 mM OA solution in the absence or presence of 200 nM Ex-4 (E7144-.1MG, Tocris, Minneapolis, MN, USA). The optimal concentrations of OA and Ex-4 we used were determined in our previous paper [18]. Briefly, the Ex-4 concentration was determined by a dose–response experiment. Different concentrations (100 to 600 nM) and times (1 h to overnight) were conducted; the best results were obtained with a treatment of 200 nM for 3 h. A longer duration of Ex-4 treatment and higher concentration did not improve the outcome, likely because Ex-4 was degraded in the culture media after 3 h. For each experiment, we used a fresh aliquot of Ex-4. Cell viability was checked, and cells demonstrated viability ranging from 80 to 90% in each step of the experiment.

### 4.3. Quantification of Steatosis

As described in our previous studies, the steatosis was quantified by triglyceride measurement [18,66] using a commercial fluorometric test kit (Abcam TG Quantification Assay Kit, ab65336) and a microplate reader to detect total TGs levels (Infinite F200 Pro; Tecan, Männedorf, Switzerland). We also used imaging of lipid droplets labeled with BODIPY 493/503, which labels neutral lipids. Finally, we used the mRNA expression of three perilipin proteins that associate with the surface of lipid droplets. 

### 4.4. Total RNA Isolation

Total RNA, including miRNAs, was extracted using a miRNeasy Mini Kit according to the manufacturer’s instructions. RNA concentrations were assessed using the NanoDropTM spectrophotometer (Thermo Fisher Scientific, Waltham, MA, USA). The RNA samples were immediately frozen at −80 °C until use. The RNA quality was assessed using the Agilent RNA 6000 Nano Kit (5067-1511, Agilent, CA, USA) and Agilent 2100 Bioanalyzer (Agilent Technologies) as per the manufacturer’s instructions. Our samples’ RNA integrity numbers (RIN) ranged from 8.8 to 10, indicating a high degree of RNA integrity. Small RNAs represented 20% of the total RNA, while miRNAs represented 17% of the total small RNAs. We saw no significant changes to these percentages upon treatments with OA and Ex-4. 

We used an RNA Broad-Range Assay Kit (Q10211, Invitrogen, Carlsbad, CA, USA) and Qubit 2.0 (Thermo Fisher Scientific, Waltham, MA, USA) to measure the RNA concentration and an Agilent RNA 6000 Nano Kit (5067-1511, Agilent, CA, USA) and Agilent 2100 Bioanalyzer (Agilent Technologies, Santa Clara, CA, USA) to assess the RNA quality, as per the manufacturer’s instructions. RNA concentrations ranged from 266.4 to 1680.8 ng/uL. RNA with a 260/230 nm absorbance ratio of >1.8 and 260/280 nm absorbance ratio > 1.8 was used for subsequent experiments on the NanoString nCounter platform. 

### 4.5. NanoString Analysis Platform and miRNA Profiling

Total RNA samples extracted from the control and treated cells were analyzed using a highly multiplexed assay to detect specific miRNAs. This test was performed on Nanostring’s nCounter (Nanostring Technologies, Seattle, WA, USA) platform using a human V3 miRNA panel that covered 799 highly curated human miRNAs, according to the manufacturer’s instructions. Briefly, 100 ng of total RNA was annealed, followed by ligation, and finally by hybridization. Hybridization was performed using reporter and capture probes at 65 °C, followed by purification by removing excess probes using the nCounter Prep Station. MiRNAs expression data were generated on the nCounter Digital Analyzer. Before data analysis, the assay’s technical performance was assessed by checking the data quality control using nSolver analysis software 4.0. To verify the sample integrity, quality, and background, positive and negative proprietary spike-in controls, hybridization controls, and ligation-specific controls were used. Five housekeeping genes (RPLP0, GAPDH, ACTB, RPL19, and B2M) were used. In order to normalize the nCounter data, the following calculation was used for each sample: Normalized count or miRNA = (raw count of miRNA/Total count of housekeeping genes) × 10,000. Stringent normalization of miRNA data was achieved by eliminating digital counts below three. A comparison of miRNA expression between the different groups was performed, and heatmaps and ratio tables with statistically significant differences were generated. 

### 4.6. Quantification Reverse Transcriptase PCR (qRT-PCR)

For hsa-miR-122a, we used the miScript II RT Kit with HiSpec Buffer (cat. No. 218160, Qiagen, Germantown, MD, USA) to reverse transcribe 1 μg of RNA into cDNA. q-PCR was performed on the QuantStudio 6 FlexTMTM qPCR (Applied Biosystems, Foster City, CA, USA) using miScript SYBR Green PCR Kit (cat. No. 218073, Qiagen, Germantown, MD, USA), and relative levels of hsa-miR-122a were determined from the respective CT values normalized against SNORD95-11 transcript levels.

For miR-4488, miR-651-5p, miR-345, miR-379-5p, and miR-1246, miRCURY^®^LNA^®^ RT Kit (Cat. No. 339340, Qiagen, Germantown, MD, USA). The quantitative PCR was performed using miRCURY LNA SYBR Green PCR Kit (200) (Cat. No. 339345, Qiagen, Germantown, MD, USA). Relative expression of miR-4488, miR-651-5p, miR-345, miR-379-5p, and miR-1246 were normalized against SNORD48.

### 4.7. Statistical Analyses

We performed all statistical analysis and graphing using GraphPad Prism 9.0 software (GraphPad Prism v9, La Jolla, CA, USA). For q-PCR, we used a t-test analysis to evaluate the significance between the mean values of different experimental groups. All values are expressed as the mean ± SE (*n* = 3). Ns: not significant, * *p* < 0.05, ** *p* < 0.01, *** *p* < 0.001. The experiment was performed in triplicate.

### 4.8. Functional, Biological Pathway, and Statistical Analysis

For differential expression of miRNAs (DEmiRs), we used stringent criteria consisting of a fold change (FC) >2 and a false discovery rate (FDR) < 0.05. The significant DEmiRs were subjected to Ingenuity Pathway Analysis (IPA) (QIAGEN Redwood City, CA, USA) to identify specific networks and pathways and STRING (https://string-db.org/; accessed on 5 October 2022) for protein–protein interactions. The Venn diagrams were created using Venny 2.1 (https://bioinfogp.cnb.csic.es/tools/venny/ (accessed on 9 January 2023)). We performed all statistical analysis using GraphPad Prism 9.0 software (GraphPad Prism v9, La Jolla, CA, USA). A statistically significant difference was considered at *p*-value ≤ 0.05. 

## 5. Conclusions

In conclusion, our study showed differential expression of various miRNAs between steatotic compared to control cells and between EX-4-treated compared to non-treated steatotic cells. Investigating DEmiRs may help advance our understanding of the mechanisms that underlie the beneficial effect of GLP-1R agonists on NAFLD, providing novel insights into NAFLD pathogenesis and treatment and opening new avenues for drug target discovery. Functional analyses are needed to validate the present observations and better define the role of the DEmiRs in the development of steatosis and the positive effect of GLP-1R agonists on NAFLD. The main limitation of the present study is the use of the HepG2 cell line instead of primary hepatocytes. In addition, our findings need to be validated in other cell lines to assess the reproducibility and generalizability of our results. A full in vivo examination of the DEmiRs and pathways is required in the future to validate the current findings.

## Figures and Tables

**Figure 1 ijms-24-11606-f001:**
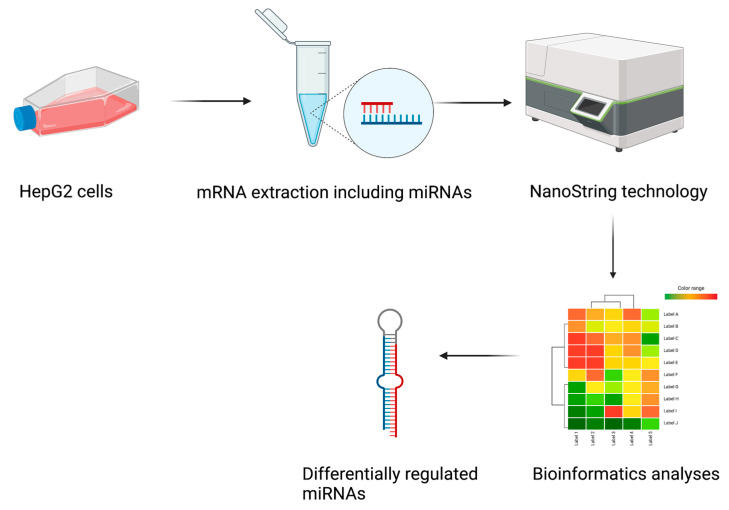
Study design. Samples from control, steatotic, and Ex-4 treated cells were collected to extract total RNA, including miRNA. The matrix for the NanoString technology was produced. Many bioinformatics tools were employed to analyze and display the collected data.

**Figure 2 ijms-24-11606-f002:**
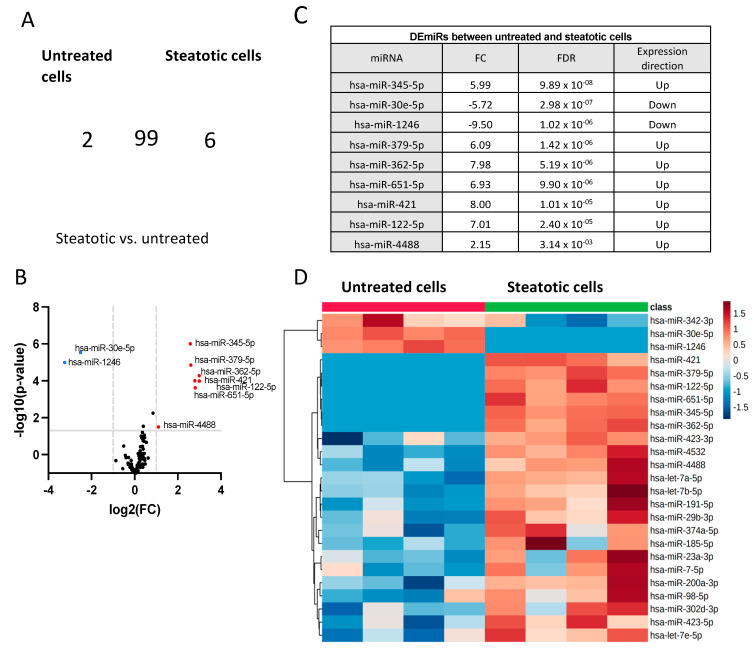
The differentially expressed miRNAs between steatotic and control cells. (**A**) Venn diagram showing the distribution of miRNAs affected by steatosis. (**B**) Volcano plot of differentially expressed miRNAs. Single miRNAs are depicted as dots. miRNAs were upregulated (red) or downregulated (blue) in steatotic compared to control cells (FC > 2 and FDR < 0.05). (**C**) List of DEmiRs between steatotic and control cells. (**D**) The hierarchical clustering heatmap of miRNA expression data from the control and steatotic cells based on a *t*-test (*p* < 0.05) (*n* = 4 for each condition). The miRNA species are shown on the right. The heatmap is based on normalized miRNA expression values from each dataset. The dendrogram shows significantly different expression levels of miRNAs among samples. Brown and blue indicate high and low expression, respectively.

**Figure 3 ijms-24-11606-f003:**
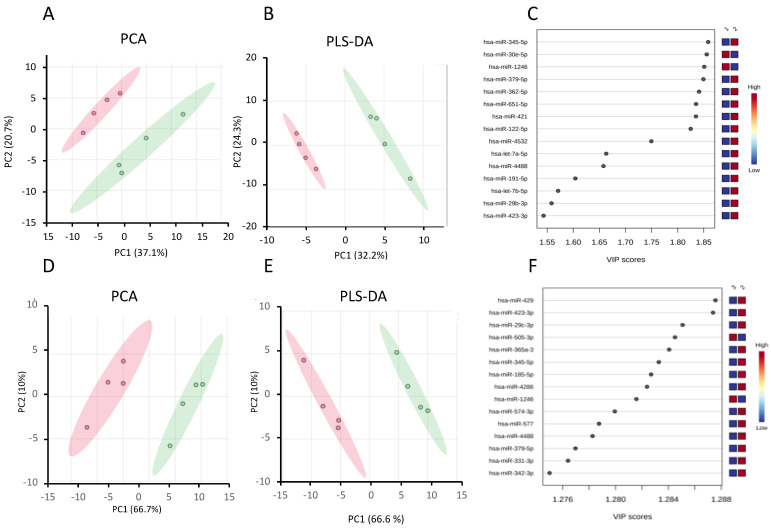
Chemometric analysis of the miRNA data set from the steatotic compared to control samples, and Ex-4-treated to steatotic samples. (**A**) The PCA score plot between steatotic and control samples. (**B**) The PLS-DA score plot between steatotic and control samples. (**C**) The important features (VIP scores) identified by PLS-DA between steatotic and control samples. The colored boxes on the right indicate the relative levels of the corresponding element in control (1) and steatotic (2) cells. The colored oval shapes in (**A**,**B**) indicate the 95% confidence regions. (**D**) The PCA score plot between the Ex-4-treated and steatotic samples. (**E**) The PLS-DA score plot between the Ex-4-treated and steatotic samples. (**F**) The important features (VIP scores) identified by PLS-DA between the Ex-4-treated and steatotic samples. The colored boxes on the right indicate the relative levels of the corresponding element in Ex-4-treated (3) and steatotic (2) cells. The colored oval shapes in (**D**,**E**) indicate the 95% confidence regions.

**Figure 4 ijms-24-11606-f004:**
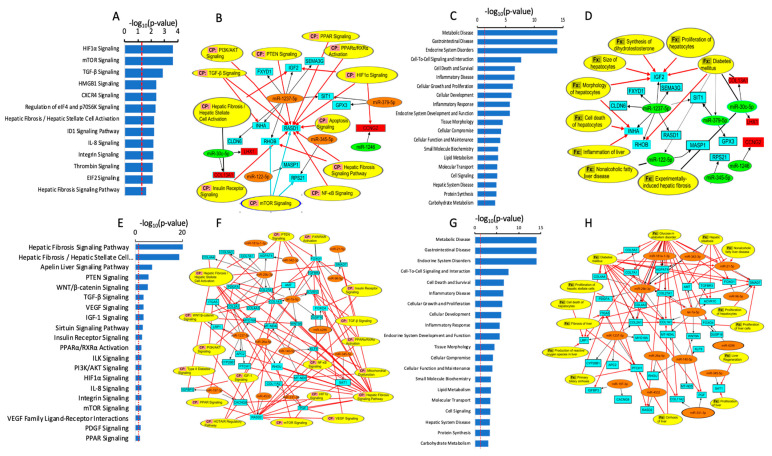
miRNA–mRNA interaction and enriched canonical pathways (CP) (**A**,**B**,**E**,**F**) and diseases and functions (Fx) (**C**,**D**,**G**,**H**) using differentially expressed miRNAs between the control cells compared to steatotic cells (**A**–**D**) and Ex-4-treated compared to steatotic samples (**E**–**H**). The canonical pathways involve at least one protein (blue (downregulated) and red (upregulated) boxes) and one miRNA (orange (upregulated) and green (downregulated) boxes). The black arrows indicate the links between the miRNAs and their target proteins, while the red arrows indicate the relationship between proteins and canonical pathways or cellular function.

**Figure 5 ijms-24-11606-f005:**
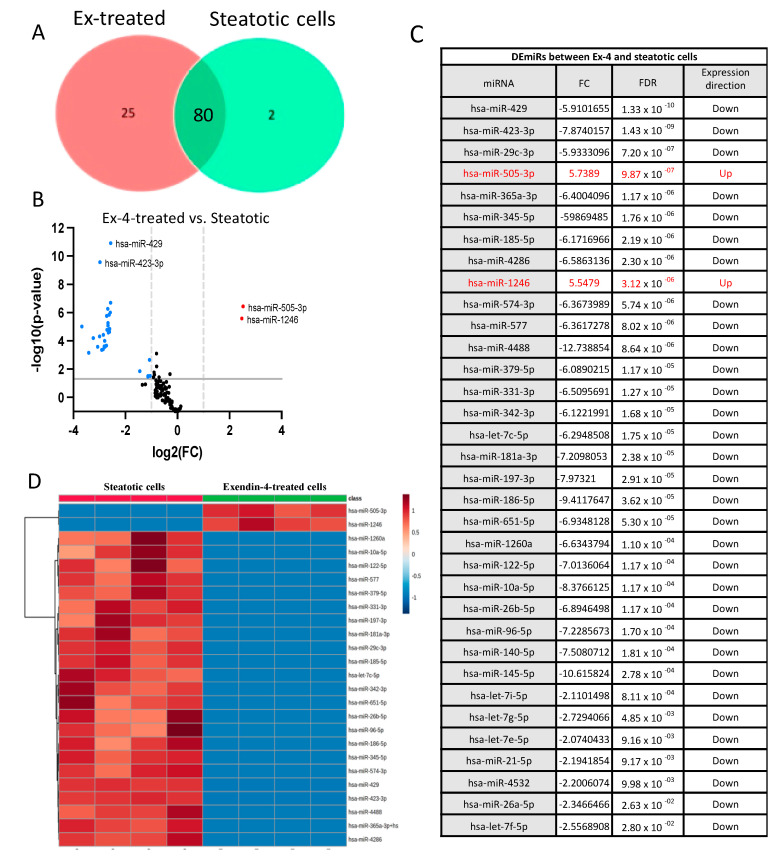
The differentially expressed miRNAs between Ex-4-treated and steatotic cells. (**A**) Venn diagram showing the distribution of miRNAs affected by Ex-4 treatment. (**B**) Volcano plot of differentially expressed miRNAs. Single miRNAs are depicted as dots. miRNAs were upregulated (red) or downregulated (blue) in Ex-4-treated compared to steatotic cells (FC > 2 and FDR < 0.05). (**C**) The list of DEmiRs between steatotic and control cells. (**D**) Heatmap of miRNA expression data from the Ex-4-treated and steatotic cells. Only the top 25 miRNAs are shown (*n* = 4 for each condition). The miRNAs species are shown on the right. The heatmap is based on the normalized miRNA expression values from each dataset. The dendrogram shows significantly different expression levels of miRNAs among the samples. Brown and blue indicate high and low expression, respectively.

**Figure 6 ijms-24-11606-f006:**
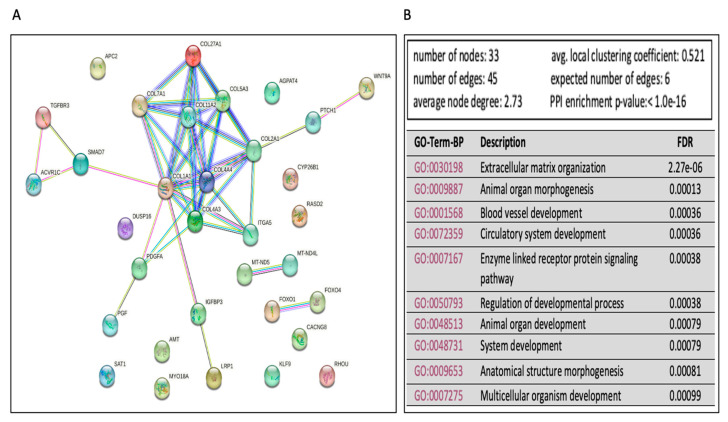
Functional enrichment analysis of mRNA associated with miRNAs dysregulated in Ex-4 treated cells versus steatotic cells. (**A**) The protein–protein interaction (PPI) network generated for the 34 gene targets of the identified DEmiRs between the Ex-4 treated and steatotic cells. Network nodes represent proteins, while the edges depict protein–protein associations. The key network statistics are also presented. (**B**) The top 10 functional enrichment annotations from Gene Ontology (GO) Biological Process are listed. STRING DataBase 11.5 was used for data analysis.

**Figure 7 ijms-24-11606-f007:**
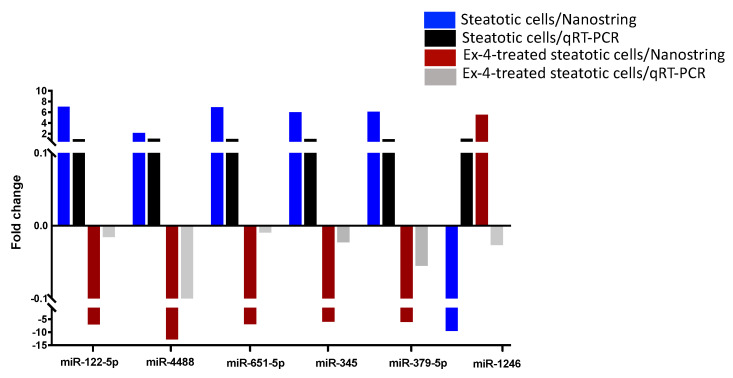
Validation of the differential expression of a set of miRNAs with qRT-PCR. Comparison of the expression of miR-122, miR-4488, miR-651, miR-345, miR-379, and miR-1246 between the steatotic cells and Ex-4-treated steatotic cells using qRT-PCR and Nanostring. The experiment was performed in triplicate.

**Table 1 ijms-24-11606-t001:** Experimentally validated targets of the differentially expressed miRNAs between steatotic and control cells.

miRNAs	Expression Direction in OA	Target mRNA *	Expression Direction in OA
hsa-miR-122-5p	Up	MASP1	Down
hsa-miR-4488	Up	CLDN6, FXYD1, IGF2, INHA, RASD1, RHOB, SEMA3G, SIT1	Down
hsa-miR-1246	Down	CCNG2	Up
hsa-miR-30e-5p	Down	COL13A1, LHX1	Up
hsa-miR-345-5p	Up	RPS21	Down
hsa-miR-379-5p	Up	GPX3, SIT1	Down

*: Target mRNA were previously identified using a transcriptomics analysis [28]. mRNA data was used to identify the gene targets of the differentially expressed miRNAs.

**Table 2 ijms-24-11606-t002:** Experimentally validated gene targets of the differentially expressed miRNAs between the Ex-4-treated and steatotic cells.

miRNAs	Expression Direction in Ex-4	Target mRNA *	Expression Direction in Ex-4
hsa-let-7c-5p	Down	ACVR1C, AMT, COL1A1, COL27A1, DUSP16, MT-ND4L, TGFBR3, WNT9A	Up
hsa-miR-4488	Down	APC2, COL2A1, CYP26B1, ITGA5, LRP1, TIAF1, PTCH1	Up
hsa-miR-140-5p	Down	KLF9, MT-ND4L, MT-ND5, RHOU, WNT9A	Up
hsa-miR-181a-3p	Down	AGPAT4	Up
hsa-miR-197-3p	Down	CACNG8, IGFBP3	Up
hsa-miR-21-5p	Down	SMAD7	Up
hsa-miR-26b-5p	Down	RHOU	Up
hsa-miR-29c-3p	Down	AGPAT4, COL1A1, COL27A1, COL2A1, COL1A1, OL1A27A1, COL2A1, COL4A3, COL4A4, COL5A5, COL7A1, PDGFA	Up
hsa-miR-331-3p	Down	COL11A2, PGF	Up
hsa-miR-342-3p	Down	AGPAT4	Up
hsa-miR-345-5p	Down	SAT1	Up
hsa-miR-4286	Down	FOXO4	Up
hsa-miR-4532	Down	RASD2	Up
hsa-miR-96-5p	Down	FOXO1, SMAD7	Up

*: Target mRNAs were previously identified using a transcriptomics analysis [28]. mRNA data were used to identify the gene targets of the differentially expressed miRNAs.

## Data Availability

All data generated or analyzed during this study are included in this published article or are available from the corresponding author by reasonable request.

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
