# Peer review of "Targeted MicroRNA Profiling Reveals That Exendin-4 Modulates the Expression of Several MicroRNAs to Reduce Steatosis in HepG2 Cells"

_ijms, 2023, doi:10.3390/ijms241411606_

Round 1

Reviewer 1 Report (Previous Reviewer 1)

The authors are to be commended for performing additional experiments. This significantly strengthens the manuscript. I have a few comments to further refine the manuscript, particularly the new qPCR data included:

Major Comments

1. Please include error bars in Figure 7, show all points and asterisks/ns, etc., highlighting the various groups comparisons.
Please also include in the legend of figure 7, the names of the statistical tests performed 

2. Please include more details regarding the qPCR validated miRNA in the results section 2.4, including fold changes and exact p values for the validated miRNA

3. Please also update the discussion with results of the qPCR analysis. For e.g., lines 389-392 could be supplemented with the fold changes and p-values from the qPCR

4. Please also briefly include in the introduction that you have performed qPCR to validate your results

Minor Comments

5. The supplemental tables (1&2) could be included in the main manuscripts. Please include a footnote informing the reader on the sources of the experimentally validated targets. (either include the original source or the methods program that generated this output).

6. In results section 2.2, It may help to include fold changes in the results

English language in manuscript is largely alright.
Copy editing should cover any remaining language corrections.

Author Response

We appreciate the reviewer's time in reading and commenting on our paper. Attached are our detailed responses to each of the issues raised.

Reviewer 2 Report (New Reviewer)

The authors explored how Exendin-4 affects microRNA expression profile in an in vitro hepatic steatosis model (HepG2 cells treated with oleic acid). In general, the paper is well written and the topic is interesting in the field.

Below are some suggestions I have.

1. Several figures and tables appear stretched (for example, Figure 6). Please properly resize the figures and tables in the manuscript for publication purpose.

2. Only one human hepatoma cell line was used in this study. Did the authors repeat the major findings of this manuscript in other human hepatoma cell line?

Quality of English Language is good.

Author Response

We appreciate the reviewer's time in reading and commenting on our paper. Attached are our detailed responses to each of the issues raised.

Round 2

Reviewer 2 Report (New Reviewer)

The revision is OK.

Quality of English Language is good.

This manuscript is a resubmission of an earlier submission. The following is a list of the peer review reports and author responses from that submission.

Round 1

Reviewer 1 Report

Dear Editor,

In this manuscript authors have investigated an in-vitro model of Non-alcoholic fatty liver disease or NAFLD, which is characterized by accumulation of fat in hepatocytes (steatosis) and is a leading cause of liver disease and transplantation. Authors have induced steatosis in the HepG2 cell line by Oleic Acid and assessed treated it with a GLP1 Receptor agonist Exendin-4 or Ex-4 which has shown utility in treating NAFLD by decreasing fat stores of hepatocytes. Authors have then compared the miRNA expression profile of control cells, steatotic cells and Ex-4 treated Steatotic cells and looked at potential mRNA targets and cellular processes that are affected by steatosis and by Ex-4 treatments.  

Overall, the manuscript is well written and is trying to address a health need that is lacking therapeutics. However, the manuscript lacks sufficient information to support its findings as well as reproduce its results. Following are my comments –

Major Comments

0. Use of nanopore restricts authors to 1/3rd of known human miRNAs. Why did the authors not do RNA sequencing?

1. Authors have presented findings from a single experimental method, namely the NanoString platform. The study does not validate its findings with an alternate method (such as qRT-PCR, ISH, Laser capture microdissection, etc). In my opinion, the authors have not provided sufficient evidence to support their claims.

2. Authors should provide information on RNA quality of their samples. What were the RIN scores? What was the percentage of miRNA or small RNA in the data? Does the % of miRNA to total RNA or miRNA to small RNA change following induction of Steatosis or treatments with Ex-4?

3. For Results section 2.2, please include data to confirm establishment of steatosis and improvement of steatosis with Ex-4 treatment, preferably imaging data, quantification of TG content and gene expression data of perlipin to confirm establishment of steatosis in the same cells that you did transcriptomics on. Please also comment on the quality of cell viability, and RNA quality control.

4. Did Ex-4 treatment affect the viability of HepG2 cells? Does Ex-4 have a dose dependent decrease in OA-induced lipid accumulation?

5. Figure 2 is being repeated in later figures in slices. Authors should show only one heatmap or only two separate heatmaps. Ideally a single heatmap of all groups should be shown. Figure 2B is redundant.

Also, it looks like the top 30 miRNAs include the 21 not detected in Ex-4 treated group. While these 21 may be interesting, they are not the most important finding, and could be moved to the supplemental.

The most crucial findings, if any would be the miRNAs that change upon steatosis induction, and if that change can be reversed by Ex-4 treatment. In my estimation, these would be miRNAs – miR-379-5p, miR-122-5p, miR-345-5p, miR-651-5p, miR-1246 and miR-4488. These would need to be validated by a second method (qPCR, ISH, etc) to confirm the nanopore findings.

Authors do a good job of discussing the NAFLD literature on these six miRNAs. I would recommend validating these 6 miRNAs, trimming the Discussion down to only to these miRNAs and focusing pathway and enrichment analyses to just these.  

6. Authors should focus the functional enrichment analysis on the subset of miRNA:mRNA pairs that IPA shortlists based on mRNAseq expression data. If any miRNA:mRNA pair from this subset changes following treatment with Ex-4 and reversing effects of steatosis, that would be a crucial finding. If authors could validate the mRNAs also by qPCR, it would strengthen the manuscript.

7. Which miRNA:mRNA pairs from Table 1 & Table 2 overlap? miRNA expression that is being reversed from EX-4 treatment => their mRNA targets should be highlighted in the main manuscript. I recommend that Tables 1 & 2 be moved to supplemental.

8. Raw Nanopore Data should be included/made publicly available.

Minor Comments

Introduction

- Has anyone investigated miRNAs in fatty liver disease? In humans or in animal models? If so, what miRNAs have been implicated? The authors should consider addressing this with relevant citations.

- On page 2, in the following statements, I would recommend citing primary sources rather than is citing reviews.

Since their discovery in Caenorhabditis elegans in 1993 [20], these short non-coding RNAs have been linked to various biological processes, including development, metabolic regulation, aging, and disease progression [21]. Notably, miR- NAs have recently been found to play a significant role in lipid metabolism, inflammation, cell death, and tissue development, all of which significantly contribute to the risk of NAFLD [22, 23].”

- On page 2, authors should cite original studies where Oleic Acid was used to develop an in-vitro model of steatosis

Materials & Methods

- Does ATCC recommend passaging HepG2 cells up to 25 times?

- What was the maximum duration of storage of OA at 4 degrees? How long is OA stable in solution at 4 degrees?

- How was Ex-4 formulated? Authors should include a sentence or two elaborating on why these doses were selected.

- The authors should elaborate on how the miRNA data was analyzed in graph pad. What criteria did the authors employ to filter & normalize the miRNA detected. What statistical tests did the authors perform in graph pad prism? How did they correct for multiple comparisons? What were some of the highest expressed miRNAs in each group? What was the level of expression of the DE miRNAs?

Results

- Please Note, the heatmap is a type of unsupervised hierarchical clustering.

- If Figure 1 was prepared using Biorender, please cite it.

- In Figure 2, Venn Diagram is missing the circles. Please re-upload this image.

- For Heatmaps, I recommend using colors other than red and blue for labelling the groups as heatmap also utilized red and blue

- In Figure 2B, Figure 4, Figure 5, Figure 6, I recommend increasing font of the miRNAs for readability

- Please include n/a in the venn diagram in the instance of the miRNA that was not detected at all in a given group

- On page 6, This sentence makes no sense to me.

This section may be divided by subheadings. It should provide a concise and precise description of the experimental results, their interpretation, as well as the experimental conclusions that can be drawn.’

- On page 7, Is this supposed to be ‘b)’?

a) Differentially expressed miRNAs in Ex-4-treated versus steatotic cells’

- Please increase the font of axes labels, legend labels, axes and legend titles for figure 4. There are mistakes in the labelling of the figure 4

Author Response

We appreciate the reviewer's time in reading and commenting on our paper. Attached our detailed responses to each of the issues raised.

Reviewer 2 Report

Title: Targeted microRNA profiling reveals that Exendin-4 modulates the expression of several microRNAs to reduce steatosis in HepG2 cells

Non-alcoholic fatty liver is a reversible non-neoplastic liver disease affecting most of the human population due to altered unhealthy food habits and sedentary lifestyles. Many complex cellular pathological events in the development of non-alcoholic fatty liver appear to be regulated by multiple factors. microRNA is a small non-coding RNA known to regulate many genes functions post-transcriptionally. In this current research work, the author has studied the expression of various microRNA in the in vitro model of steatosis using HepG2 cells. This research highlights the spectrum of microRNA that are up and downregulated in invitro steatosis conditions supporting the pathophysiology of fatty liver and the beneficial effect of Exendin-4 via regulating these microRNA to prevent the development of the non-alcoholic fatty liver.

Comments

1.      The author is expected to use the term “Control” to denote the healthy cells instead of “Untreated cells”.

2.      Does the Exendin-4 treatment to the healthy HepG2 cells alter the microRNA expression? It is curious to know the microRNA profile of healthy HepG2 cells treated with Glucagon-Like Peptide-1 Receptor Agonists to state that the Exendin-4 treatment selectively alters the microRNA expression in steatotic versus control HepG2 as a promising therapeutic tool.

3.      The Venn diagram of figure 2 needs to be clarified.

4.      Line number 373 needs to be revised for β-catenin.

5.      The author has found specific microRNA up and downregulated in the steatotic HepG2 cells compared with control HepG2 cells, whereas the Exendin-4 treated steatotic HepG2 cells had certain additional microRNA profiles compared with steatotic HepG2 cells. The author needs to provide a more critical discussion on such massive alteration in microRNA profile for a clear mechanistic understanding of the therapeutic potential of Exendin-4 via altering microRNA profile.

6.      The author is expected to update the manuscript with additional in-vitro data like oil red O staining images of HepG2 cells under all the experimental conditions to understand the degree of fatty changes.

7.      Different microRNA was altered in the steatotic, and Exendin-4 treated steatotic HepG2 targeting inflammation and liver fibrosis. These events are associated with other liver cells populations like Kupffer cells and hepatic stellate cells. The author needs to provide a detailed critical discussion of these findings.

Author Response

We are appreciative of the reviewer's time and effort in reviewing our manuscript. To the best of our ability, we have responded to each of the concerns raised below.

Round 2

Reviewer 1 Report

Authors have improved the manuscript in the revised submission.

However, qPCR validation is critical. Authors should take their time to obtain the materials and run the validation. If authors need more than 10 days to complete these experiments, IJMS can and should grant them sufficient time to complete the experiments.  

Author Response

Please find attached our replay to the reviewers in the cover lettre.

Reviewer 2 Report

Title: Targeted microRNA profiling reveals that Exendin-4 modulates the expression of several microRNAs to reduce steatosis in HepG2 cells.

The author has well revised the manuscript. However, it is very difficult to conclude that Exendin-4 selectively alters the micro RNA profile in steatotic HepG2 cells because it lacks the actual control for Exendin-4 to state that Exendin-4 does not affect the normal microRNA profile of control HepG2 cells. 

Author Response

Please find attached our replay to the reviewers in the cover lettre 
